# Peptide Mapping for Sequence Confirmation of Therapeutic Proteins and Recombinant Vaccine Antigens by High-Resolution Mass Spectrometry: Software Limitations, Pitfalls, and Lessons Learned

**DOI:** 10.3390/ijms26209962

**Published:** 2025-10-13

**Authors:** Mateusz Dobrowolski, Małgorzata Urbaniak, Tadeusz Pietrucha

**Affiliations:** 1Research and Development Department, Mabion S.A., Langiewicza 60, 95-050 Konstantynów Łódzki, Poland; 2Department of Medical Biotechnology, Medical University of Lodz, Żeligowskiego 7/9, 90-752 Łódź, Poland

**Keywords:** antibody, peptide mapping, SARS-CoV-2, electrospray ionization mass spectrometry, sequence confirmation, false positives, orbitrap, biosimilar

## Abstract

Peptide mapping is a well-established method for confirming the identity of therapeutic proteins as part of batch release testing and product characterization for regulatory filings. Traditionally based on enzymatic digestion followed by reversed-phase liquid chromatography and UV detection, the method has evolved with technological advancements to incorporate mass spectrometry (MS), enabling more detailed structural insights. Residue-level confirmation of amino acid sequences requires MS/MS fragmentation, which produces large amounts of data that must be processed using specialized software. In regulated environments, the use of academic algorithms is often limited by validation requirements, making it necessary to rely on commercially approved tools, although their built-in scoring systems have limitations that can affect sequence assignment accuracy. Here, we present representative examples of incorrect peptide assignments generated by commercial software. In antibody sequence analysis, misidentifications resulted from isobaric and near-isobaric dipeptides (e.g., SA vs. GT). Additional examples from the analysis of SARS-CoV-2 spike protein variants revealed software-induced artifacts, including artificial succinylation of aspartic acid residues to compensate for sequence mismatches, and incorrect deamidation site assignments due to misinterpretation of isotopic peaks. These findings underscore the necessity for expert manual review of MS/MS data, even when using validated commercial platforms, and highlight the molecular challenges in distinguishing true sequence variants from software-driven artifacts.

## 1. Introduction

The advancement of molecular biology and genetic engineering in the 1970s marked a pivotal moment, leading to the development of the first recombinant protein, human insulin [1]. Its approval by the U.S. Food and Drug Administration (FDA) in 1982 marked a significant milestone in biotechnology and medicine [2], laying the groundwork for the subsequent creation of a diverse array of recombinant therapeutic proteins. Since then, numerous recombinant proteins, including growth hormones, clotting factors, and monoclonal antibodies, have been successfully developed and introduced to the market for various medical applications [3,4].

The amino acid sequence plays a crucial role in the biological activity, conformational stability, and immunogenicity of a biological drug. It has been demonstrated that a single amino acid substitution in the protein’s binding region can significantly reduce or even abolish the interaction with its target, resulting in diminished or no therapeutic effect for therapeutic proteins [5,6,7]. Efficacy can be indirectly impacted by changes in protein structure influenced by mutations in the amino acid sequence [8,9]. Furthermore, alterations in protein structure may expose normally buried epitopes or promote aggregation, leading to an immunogenic reaction [10,11,12,13,14,15,16]. These reasons underscore the regulatory requirements, enforced by agencies such as the European Medicines Agency (EMA), Pharmaceuticals and Medical Devices Agency (PMDA), or FDA, for confirming the sequence of therapeutic proteins [17,18,19,20].

In 2006, the first biosimilar drug received authorization for use in the European Union, marking the onset of the biosimilar era and paving the way for the development and approval of subsequent biosimilar products. To ensure comparable efficacy and safety to the originator, a demonstration of an identical amino acid sequence is imperative [21,22,23].

For decades, Edman degradation was the preferred method for determining the amino acid sequences of proteins [24,25]. However, limitations in the length of sequenced peptides made the method both cost and time consuming. Additionally, Edman degradation cannot confirm post-translationally modified amino acids, such as N-terminal pyroglutamate formed by glutamine cyclization, common in therapeutic proteins [26,27,28]. Due to these limitations, tandem mass spectrometry has increasingly complemented or replaced Edman sequencing in recent years. Sequencing by LC-MS/MS is now considered the current standard for amino acid sequence analysis due to the high sensitivity, resolution, and throughput of modern mass spectrometry technologies [29,30,31,32,33].

Therapeutic proteins, such as monoclonal antibodies, are typically large and structurally complex molecules. Consequently, the bottom-up approach is the preferred method for amino acid sequence confirmation, where the proteins are typically enzymatically digested, and the obtained peptides are separated using liquid chromatography and analyzed with mass spectrometer. Sample preparation is a multi-step process and plays a critical role in achieving a high level of sequence coverage. Numerous publications have addressed various aspects of sample preparation and analysis in this context [34,35,36]. However, despite its comparable importance to data generation, the interpretation of peptide mapping data remains relatively underexplored in the literature. Accurate assignment of LC-MS/MS data, even with software assistance, continues to pose significant challenges and is essential for unambiguous sequence confirmation.

Over the years of working with the protein sequence confirmation, our group has encountered various situations, demonstrating that even applying strict criteria for MS/MS data evaluation may be still insufficient. Extensive experience in characterizing numerous monoclonal antibody therapeutics (including rituximab, cetuximab, and denosumab), as well as recombinant SARS-CoV-2 spike protein variants, has led us to develop a set of practical rules aimed at improving the reliability of data interpretation in protein sequence analysis via peptide mapping. Using the examples of denosumab, cetuximab, and SARS-CoV-2 variant sequence confirmation, we highlight the importance of critically assessing peptide identifications generated by commercial peptide mapping software such as BioPharma Finder 5.1, including its built-in scoring parameters. All examples discussed herein are derived from in-house developed and qualified or validated LC-MS/MS methods. Although we are aware of advanced academic algorithms for peptide-spectrum match (PSM) scoring and post-translational modification (PTM) localization [37,38,39,40,41,42,43,44], these were not considered in our study, as they are primarily designed for proteomics research rather than residue-by-residue sequence confirmation, and they are often difficult or impossible to validate for use in a regulated quality control (QC) environment, where software must comply with regulatory requirements [45,46].

The root cause of software misidentifications in the first two examples presented in the Section 2 is the presence of dipeptides with similar or identical masses in the peptide fragments. With the 20 natural amino acids, there are 190 possible dipeptide combinations (not taking into consideration the specific position of each amino acid in the dipeptide or dipeptides composed of two identical amino acids). Among these, 13 pairs of dipeptides share exactly the same elemental composition, resulting in identical mass. Moreover, when extending the analysis to include isobaric dipeptide pairs (Δm ≤ 100 ppm), it can be found that 61 dipeptides share their mass with at least 1 other dipeptide. In some cases, groups of three dipeptides were identified within a 100 ppm mass accuracy window. This represents nearly one-third of all possible dipeptide combinations. While a 100 ppm error may appear large for high-resolution instruments such as Orbitrap™ analyzers, the relative mass error becomes less significant considering that the amino acids pair constitutes only a portion of a larger peptide. The examples of correct denosumab sequence assessment will demonstrate the importance of this observation in protein sequence confirmation.

In the Section 2.2, the examples of software misinterpretation are related to the over-permissive settings regarding the number of variable modifications allowed per peptide, which may result in an excessively broad search space and false-positive identifications.

Some of the amino acids can undergo post-translational modifications (PTMs), originating during protein synthesis, but also purification, storage, or even sample preparation for analytical purposes, such as sequence confirmation [47]. Many of the modifications (e.g., deamidation) are low in abundance; therefore, along with the peptide bearing the respective PTM, the unmodified peptide should be present as well. For the greater confidence of the results, where possible, the unmodified peptides should be included in the analysis. Naturally, there are also high-abundance PTMs (e.g., pyroglutamine formation at the N-termini of proteins), which may reach nearly 100% occupancy. In those cases, the sequence can be confirmed by using PTM-containing peptides. However, the MS^2^ spectra of modification-bearing peptides should be studied thoroughly, and more than two variable modifications per peptide should be avoided.

To illustrate the possible software misidentification, examples from the method development for identity confirmation of SARS-CoV-2 spike protein variants are presented. The variants of the SARS-CoV-2 spike protein differ from the Wuhan variant by around 30 substitutions and several additions and deletions, depending on the variant. Thus, to ensure that manufactured variant is the intended one, the developed method must not produce false positives, meaning that peptides covering variant-specific mutation should not be detected in the sample of another variant. To test this, the analyzed sample of a particular variant was scanned against the sequences of other variants. If any false positives were detected, their underlying causes were investigated, and appropriate modifications to the evaluation method were introduced.

## 2. Results

### 2.1. Dipeptides with Similar or Identical Mass

As part of the sequence confirmation of denosumab in the reference medicinal product (Prolia™, Amgen Europe B.V., Breda, The Netherlands), a patent database search was conducted. Two patents containing the denosumab sequence were identified (EP1409016B1 and US9695244), with discrepancies observed between the reported sequences. Five Leu/Ile substitutions were noted but are not discussed herein. Additionally, two dipeptide substitutions were identified: ^79^LeuGlu^80^/^79^AsnLys^80^ in the light chain (sequence (1)/sequence (2); refer to Table 1) and ^101^GlyThr^102^/^101^SerAla^102^ in the heavy chain (sequence (3)/sequence (4)). To confirm the correct sequence, the collected data were analyzed against both sequence variants.

The important thing to note is that in both cases, the software identifies the same ion (in the case of the light chain peptide, the ion at *m*/*z* 990.1609, and in the case of the heavy chain peptide, the ion at *m*/*z* 1181.0365) as both sequence variants (Table 1). In the first case, the ion is identified with the same mass error due to the identical mass of dipeptide variants intended to be distinguished (Table 1). Yet, it can be observed that the average structural resolution (ASR), a metric reflecting the degree of peptide bond fragmentation (see Data analysis in the Section 4 for details) for the first sequence variant is lower and has its minimal value, meaning all peptide bonds were fragmented, and confirmed by b- and/or y-type ions (Figure 1). However, if only the results for the second sequence were available, the variant could not be excluded. Looking into the MS^2^ spectrum, it can be observed that the b_3_ ion corresponding to DPS sequence is missing (Figure 2 and Figure 3), which is reflected in the ASR value. The missing fragmentation (missing b_3_ ion) is the only basis to eliminate the second sequence variant. Similarly, in the case of the heavy chain, the software identifies the same ion as both sequence variants, again with a similar ASR and similar mass error (Table 1). As previously, the MS^2^ spectrum confirms that sequence (3) is correct. The conclusion can be drawn based on the N-termini fragmentation pattern. Because the dipeptide masses are not identical (with a mass difference of 0.0112 Da), the b_2_ and b_3_ ions were observed only in the assessment of sequence (3) (Figure 4), whereas the b_1_ to b_4_ ions were not identified in sequence (4) due to a mass error greater than 15 ppm (Figure 5 and Figure 6, Table 2).

### 2.2. The Post-Translational Modification: You Can(not) Have Too Many

In the first example of the false-positive results search, while scanning against sequence of variant A, a peptide containing a mutation specific to the variant, along with two succinimide modifications on two aspartic acids, was found (sequence (5); refer to Table 3). The same ion, while scanning against the correct variant (variant B), was identified by the software as a peptide with an aspartic acid to glycine substitution (sequence (6)) and a sodium adduct, again with an acceptable mass error due to the very similar masses of the full peptides and a good ASR value. The fragmentation patterns of both peptides were evaluated. Good coverage with b- and/or y-type ions of the shared regions of the peptides was observed, but the coverage in fragments containing only one succinimide modification was lacking (sequence (5); Figure 7), which confirms that the identification of variant B (sequence (6)) is correct (Figure 8 and Figure 9).

The final example, similar to the previous one, additionally highlights the necessity of a careful manual evaluation of MS^2^ spectra. During the search for false-positive results, the software assigned a peptide to variant C, indicating the presence of both a specific mutation and glutamine deamidation (sequence (7); Table 4). When the data were reanalyzed against the correct variant (variant D), the same ion was assigned to a peptide containing a glycine substitution characteristic of variant D (sequence (8)), along with N-terminal carbamylation. In this case, the mass difference caused by the substitution was again compensated by a different modification. However, the fragment containing the substitution was not supported by b- and y-type ion coverage (Figure 10), suggesting that sequence (8) represents the correct one (Figure 11). Notably, the deamidation of glutamine was supported by the presence of the y_8_ product ion (Figure 12). Upon closer inspection of the MS^2^ spectrum, it was determined that the y_8_ ion had been incorrectly assigned based on the second isotopic peak rather than the monoisotopic peak (Figure 13), further confirming that the assignment to variant C (sequence (7)) was a false positive.

## 3. Discussion

Peptide mapping of therapeutic proteins is a powerful tool for confirming the amino acid sequence, both during the early stages of product development and as a release test for the identity of the manufactured drug substance (DS) and drug product (DP). While technological progress has greatly advanced mass spectrometric capabilities, several challenges remain, underscoring the importance of careful data interpretation. Our observations align with previous reports emphasizing that MS^2^ spectra are vital for sequence confirmation yet prone to misinterpretation if handled without critical evaluation [37]. Even with advanced algorithms, expert manual review of spectra remains crucial to ensure reliable results.

A number of scoring algorithms and statistical models have been developed to improve PSM confidence. Examples include Mascot’s probability-based scoring [38], Andromeda using posterior error probabilities [39], and MS-GF+, which calculates E-values for PSMs [40]. Additional statistical tools such as PeptideProphet [41] and the target–decoy strategy [42] are used to estimate the false discovery rate, while machine learning-based re-scoring tools such as Percolator further enhance separation between correct and incorrect assignments [43]. Dedicated tools for PTM site localization such as PTMProphet estimate the probability of modification sites and help control the false localization rate [44]. These developments underline the strong algorithmic framework that supports proteomics studies today.

However, these methods were developed primarily for broad proteomics-based identification rather than complete sequence confirmation and cannot fully eliminate the risk of false positives. Therefore, it is crucial to understand how the software operates and to set appropriate search parameters. As demonstrated with BioPharma Finder 5.1, over-permissive settings, such as allowing too many variable modifications on a single peptide, may artificially expand the search space, leading to false-positive identifications. Built-in quality metrics, such as confidence scores, should also be interpreted with caution, as even misidentified peptides may receive the highest score. Practical experience and rigorous parameter selection thus remain indispensable for high-confidence confirmation.

Complementary fragmentation methods, particularly electron transfer dissociation (ETD) and the hybrid EThcD approach that combines ETD with higher-energy collision-induced dissociation (HCD), offer substantial advantages for improving sequence confirmation workflows [48,49,50]. Unlike traditional HCD, which primarily produces b- and y-type ions, EThcD yields both c- and z-type fragments from ETD and b- and y-type ions from HCD, resulting in more comprehensive fragmentation patterns. This enhanced ion diversity increases the likelihood of full sequence coverage and is especially useful for confirming the position of labile post-translational modifications and resolving ambiguities in regions with incomplete fragmentation. EThcD has also proven helpful in analyzing isobaric or isomeric dipeptides, where the generation of complementary fragment ions can support the correct localization of these residues within the reference sequence, even if they are not directly distinguishable by mass. While highly promising, the practical use of ETD/EThcD in regulated QC environments remains limited by longer acquisition times per spectrum and the complexity of method validation. Nevertheless, the growing evidence for its utility in residue-level sequence confirmation suggests that these technologies should be considered for integration into critical workflows where confident discrimination of sequence variants, including isobaric dipeptides, is required.

It is also important to distinguish the scope of peptide mapping-based sequence confirmation from related fields, such as de novo sequencing. While peptide mapping relies on matching to a known reference sequence and is best suited for confirming expected sequences and detecting known PTMs, its reliability critically depends on the accuracy of the reference sequence itself. An incorrectly reported reference sequence of cetuximab has been described [51], which was subsequently confirmed using the method described herein, yet the incorrect version remains available in some online databases. By contrast, de novo methods aim to reconstruct peptide sequences directly from MS^2^ spectra without prior sequence knowledge. In particular, Yefremova et al. showed that a combined top-down and bottom-up de novo approach could reveal unexpected sequence extensions and PTMs [52]. Our work, however, focuses on sequence confirmation rather than sequence discovery, which is especially relevant for biosimilar development and regulatory use.

Finally, algorithms commonly applied in academic proteomics may not be directly applicable in regulated QC environments. In Good Practice (GxP) settings, every computerized system used for data acquisition or analysis must undergo formal validation to ensure compliance with regulatory requirements [45,46]. While such research algorithms can inspire methodological advances, their practical implementation in QC would require formal validation, which in many cases may be difficult or not feasible.

## 4. Materials and Methods

### 4.1. Materials

The denosumab (Prolia™, Amgen Europe B.V., Breda, The Netherlands) and cetuximab (Erbitux, Merck KGaA, Darmstadt, Germany) were commercially sourced. The recombinant SARS-CoV-2 spike protein variants were provided by Novavax Inc. All buffers and solutions were prepared in accordance with internal procedures, fulfilling the GMP standards with use of the following: formic acid 98–100% for LC-MS (Merck KGaA, Darmstadt, Germany); Trisodium Citrate Dihydrate (POCH, Gliwice, Poland); Citric acid monohydrate, Urea, Hydrochloric acid and Acetic acid 100% (Chempur, Piekary Slaskie, Poland); Acetonitrile LC-MS (VWR International, Radnor, PA, USA); Tris(2-carboxyethyl)phosphine hydrochloride and *N*-Ethylmaleimide (Sigma-Aldrich, St. Louis, MO, USA); and Tris (BioShop, Burlington, ON, Canada) Guanidine Hydrochloride and Pierce LTQ Velos ESI Positive Ion Calibration Solution (Thermo Scientific, Rockford, IL, USA). The proteases Trypsin Gold, Mass Spectrometry Grade, and Chymotrypsin were purchased from Promega Corporation (Madison, WI, USA).

### 4.2. Sample Preparation

The standard bottom-up approach [36] with several fit-for-purpose modifications was used. In the developed standard operating procedure, the studied protein (100–125 µg) was first denatured using a high concentration of chaotropic salt (6 M, Guanidine Hydrochloride), followed by the reduction of disulfide bridges with TCEP (Tris (2-carboxyethyl) phosphine hydrochloride) with a final concentration of 5 mM. Reassembly of disulfide bridges was prevented by alkylation of free cysteines with NEM (*N*-Ethylmaleimide) with a final concentration 5 mM. Then, the Guanidine Hydrochloride and remaining reagents were removed from the solution via ultrafiltration. The digestion was performed under mild denaturing conditions (1 M Urea) using Trypsin or Chymotrypsin for 16 h at 37 °C, with an enzyme-to-sample mass ratio of 1:59.

### 4.3. Sample Analysis

The analyses were performed using a Ultra-High-Performance Liquid Chromatography (UHPLC) system (Ultimate 3000, ThermoFisher Scientific, Germering, Germany) combined with a Q-Orbitrap™ mass spectrometer (Q Exactive Plus, ThermoFisher Scientific, Bremen, Germany). The separation of a 20–45 µg peptide mixture was conducted on an RP-C18 column (Acclaim RSLC 120 C18, 2.2 μm, 120 Å, 2.1 × 250 mm, ThermoFisher Scientific) with a 30-minute-long gradient starting from 2% acetonitrile (ACN) to 45% ACN in water with 0.1% formic acid addition under a flow rate of 0.3 mL/min. Eluted peptides were measured using the ddMS^2^ method, where for each full scan (MS^1^), 6 ions with the highest intensity were selected for HCD fragmentation and MS^2^ spectrum acquisition.

### 4.4. Data Analysis

Data analysis was performed using BioPharma Finder 5.1 software (ThermoFisher Scientific). The acquired data were searched for the theoretically possible ions corresponding to potential peptides generated based on the provided reference protein sequence. The software gives several parameters for peptide identification evaluation, e.g., the mass error (Δppm), the MS signal area or identified amino acid modification for MS^1^, and also parameters for the MS^2^, e.g., confidence score and ASR. Notably, the confidence score and ASR are built-in, software-specific metrics of BioPharma Finder and are not directly comparable with the scoring systems used in other peptide identification platforms. The confidence score reflects how closely the predicted and experimental MS/MS spectra match, expressed on a scale from 0 to 100%. The ASR is calculated as the number of amino acids divided by the number of peptide bonds for which product ions were detected plus one (Equation (1)), thereby providing a measure of the average sequence resolution available for a given peptide.(1)$$ASR=\frac{\mathrm{Number}\;\mathrm{of}\;\mathrm{Amino}\;\mathrm{Acids}}{\mathrm{Number}\;\mathrm{of}\;\mathrm{Bonds}\;\mathrm{Found}+1}.$$

The acceptance criterium for the peptide ion assignment by the software in the used standard operating procedure was |Δppm| ≤ 10 ppm in MS^1^ and |Δppm| ≤ 15 ppm for product ions in MS^2^, with a confidence score ≥ 95%. The ASR highly depends on the peptide length, especially considering the peptides shorter than 10 amino acids; thus, it is beneficial to establish different acceptance criteria depending on the length or mass of the peptide to ensure a reasonable level of peptide sequence coverage in b- and/or y-type ions. An ASR cutoff corresponding to a minimum of 40% sequence coverage by b- and/or y-type ions was applied.

## 5. Conclusions

Peptide mapping is a robust method for protein sequence confirmation, but the large amount of data generated during a single ddMS^2^ run makes reliable interpretation challenging. Software tools are necessary to process such datasets, yet the expertise of an experienced scientist plays a key role. Typical pitfalls include misinterpretation of isobaric dipeptides, inappropriate introduction of modifications by software, or errors in monoisotopic peak assignment. To improve reliability, sequence confirmation should prioritize unmodified peptides or those with highly abundant modifications, with close inspection of fragmentation patterns near the modification site, and must always rely on a reference sequence from a reliable source. With this work, we aimed to share practical observations that may support residue-by-residue sequence confirmation, particularly in regulated GxP environments where algorithmic tools cannot always be applied.

## Figures and Tables

**Figure 1 ijms-26-09962-f001:**
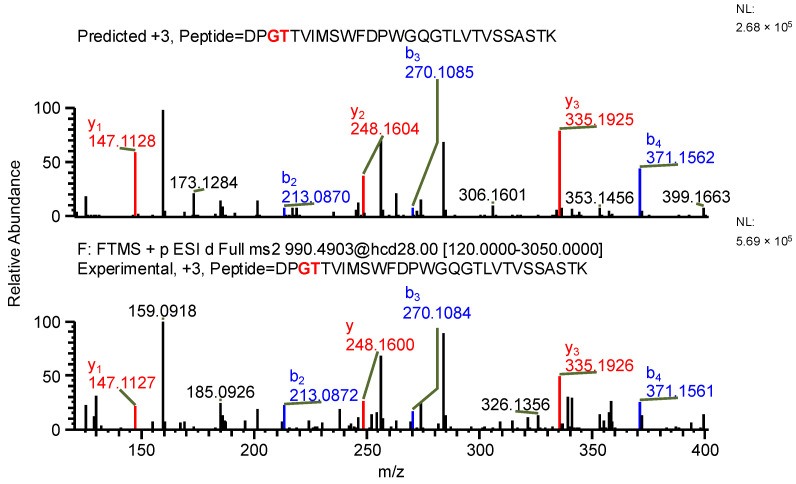
MS^2^ product ion spectrum of the precursor ion at *m*/*z* 990.1609. The peak assignment was performed using sequence (1). Top: software-predicted spectrum based on the peptide sequence; bottom: experimental spectrum. The b- and y-type ions are marked in blue and red, respectively.

**Figure 2 ijms-26-09962-f002:**
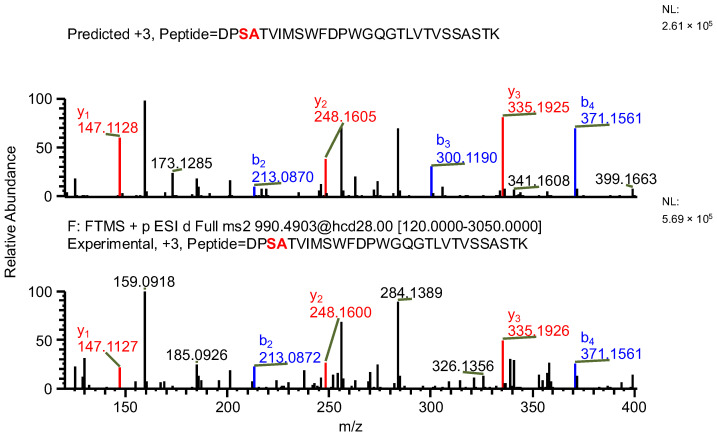
MS^2^ product ion spectrum of the precursor ion at *m*/*z* 990.1609. The peak assignment was performed using sequence (2). Top: software-predicted spectrum based on the peptide sequence; bottom: experimental spectrum. The b- and y-type ions are marked in blue and red, respectively.

**Figure 3 ijms-26-09962-f003:**
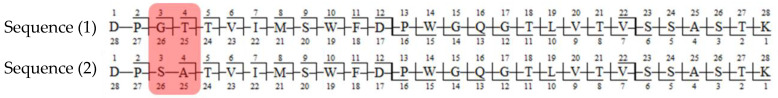
Fragmentation patterns of correctly (sequence (1)) and incorrectly (sequence (2)) assigned peptides in denosumab sequence. The substitutions are marked in red.

**Figure 4 ijms-26-09962-f004:**
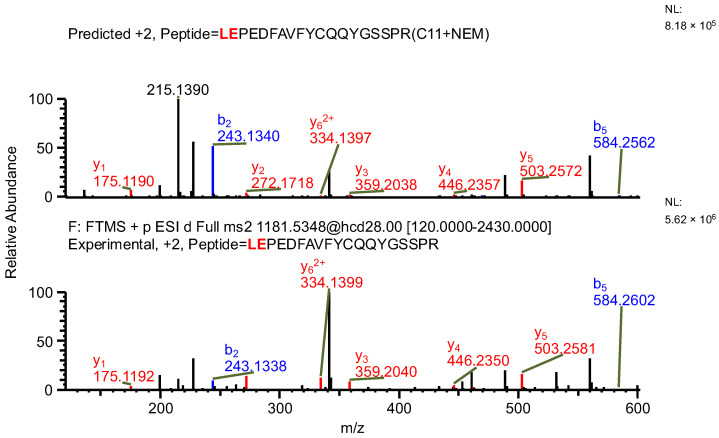
MS^2^ product ion spectrum of the precursor ion at *m*/*z* 1181.0365. The peak assignment was performed using sequence (3). Top: software-predicted spectrum based on the peptide sequence; bottom: experimental spectrum. The b- and y-type ions are marked in blue and red, respectively.

**Figure 5 ijms-26-09962-f005:**
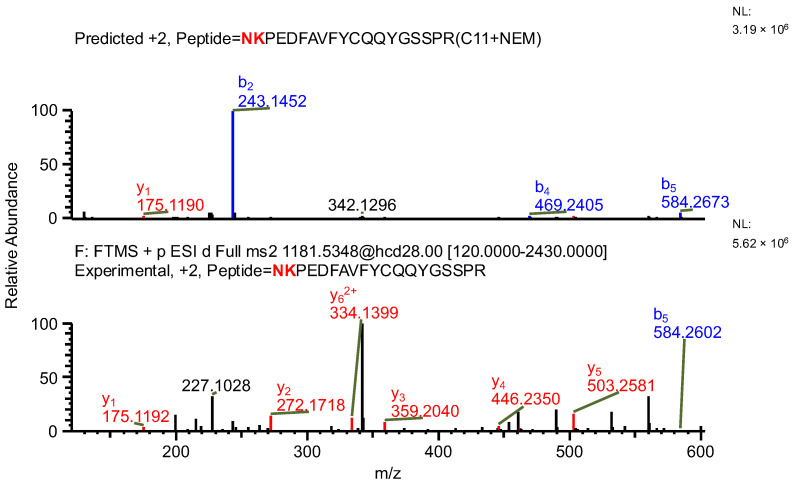
MS^2^ product ion spectrum of the precursor ion at *m*/*z* 1181.0365. The peak assignment was performed using sequence (4). Top: software-predicted spectrum based on the peptide sequence; bottom: experimental spectrum. The b- and y-type ions are marked in blue and red, respectively.

**Figure 6 ijms-26-09962-f006:**
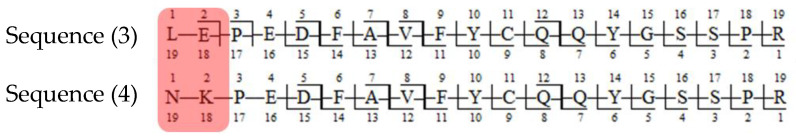
Fragmentation patterns of correctly (sequence (3)) and incorrectly (sequence (4)) assigned peptides in denosumab sequence. The substitutions are marked in red.

**Figure 7 ijms-26-09962-f007:**
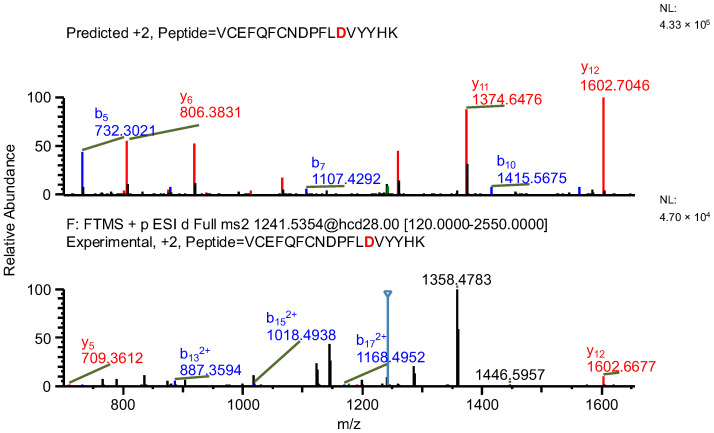
MS^2^ product ion spectrum of the precursor ion at *m*/*z* 1241.0356. The peak assignment was performed using sequence (5). Top: software-predicted spectrum based on the peptide sequence; bottom: experimental spectrum. The b- and y-type ions are marked in blue and red, respectively.

**Figure 8 ijms-26-09962-f008:**
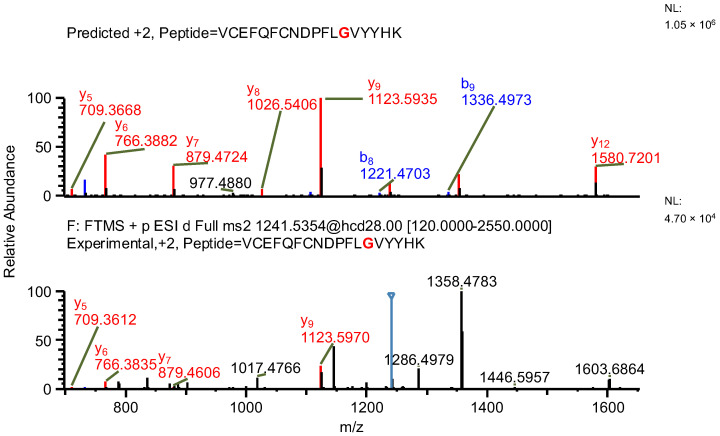
MS^2^ product ion spectrum of the precursor ion at *m*/*z* 1241.0356. The peak assignment was performed using sequence (6). Top: software-predicted spectrum based on the peptide sequence; bottom: experimental spectrum. The b- and y-type ions are marked in blue and red, respectively.

**Figure 9 ijms-26-09962-f009:**
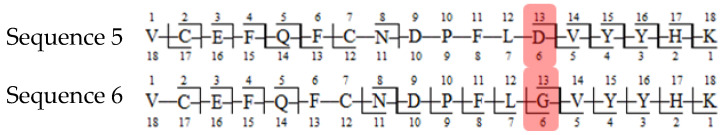
Fragmentation patterns of correctly identified peptide (sequence (6), variant B) and false-positive result (sequence (5), variant A). The substitutions are marked in red.

**Figure 10 ijms-26-09962-f010:**
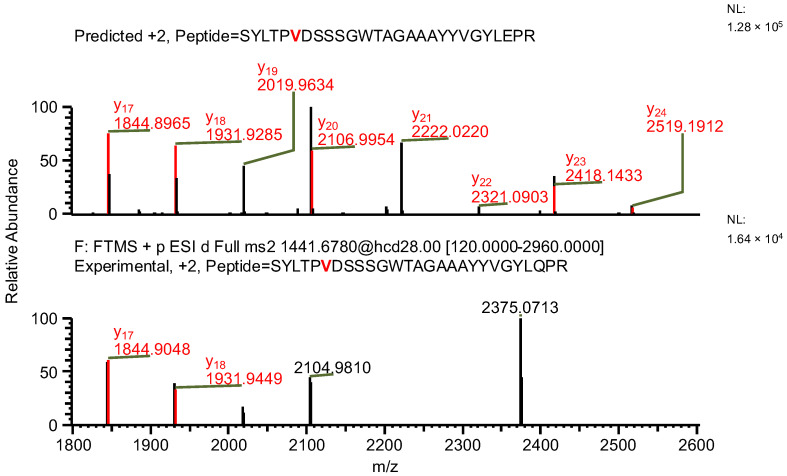
MS^2^ product ion spectrum of the precursor ion at *m*/*z* 1441.1763. The peak assignment was performed using sequence (7). Top: software-predicted spectrum based on the peptide sequence; bottom: experimental spectrum. The y-type ions are marked in red.

**Figure 11 ijms-26-09962-f011:**
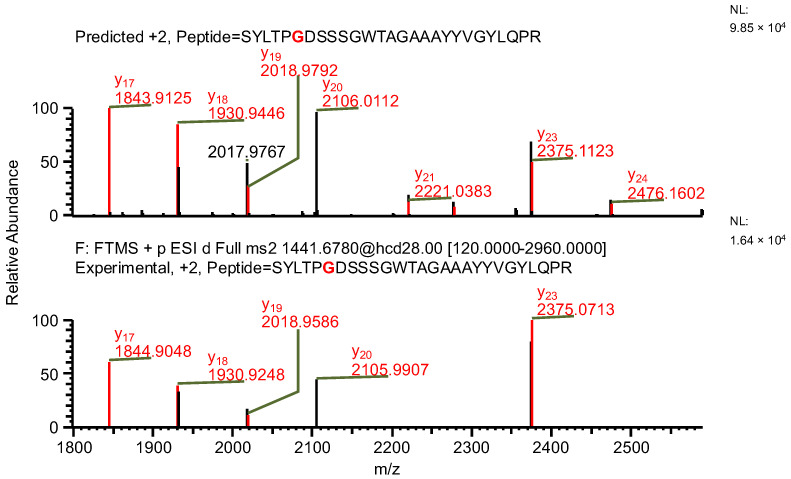
MS^2^ product ion spectrum of the precursor ion at *m*/*z* 1441.68. The peak assignment was performed using sequence (8). Top: software-predicted spectrum based on the peptide sequence; bottom: experimental spectrum. The y-type ions are marked in red.

**Figure 12 ijms-26-09962-f012:**
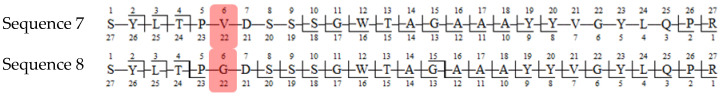
Fragmentation patterns of correctly identified peptide (sequence (8), variant D) and false-positive result (sequence (7), variant C). The substitutions are marked in red.

**Figure 13 ijms-26-09962-f013:**
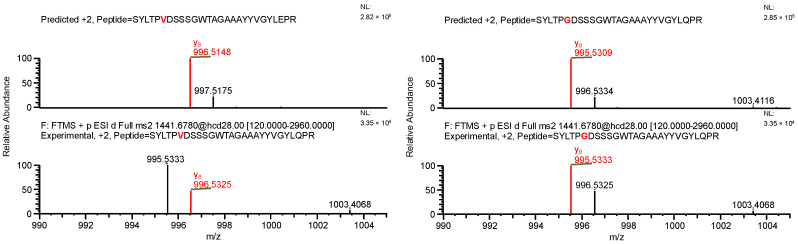
MS^2^ product ion spectra of the precursor ion at *m*/*z* 1441.1763. Top: software-predicted spectra based on the peptide sequence; bottom: experimental spectra. Left: sequence (7), variant C; right: sequence (8), variant D. The y_8_ fragment ion, highlighted in red, is shown in the zoomed region.

**Table 1 ijms-26-09962-t001:** The comparison of MS parameters of correctly and incorrectly assigned peptides in denosumab sequence.

mAb Chain	Light		Heavy	
**Sequence no.**	(1)	(2)	(3)	(4)
**Peptide sequence**	DPGTTVIMSWFDPWGQGTLVTVSSASTK	DPSATVIMSWFDPWGQGTLVTVSSASTK	LEPEDFAVFYCQQYGSSPR	NKPEDFAVFYCQQYGSSPR
**Mass accuracy [ppm]**	9.61	9.61	6.95	2.19
**Best ASR**	1.0	1.1	1.1	1.2
**Confidence score [%]**	100	100	100	100
**Retention time [min]**	31.33	31.33	26.46	26.46
** *m* ** **/*z***	990.1609	990.1609	1181.0365	1181.0365
**Experimental monoisotopic mass [Da]**	2967.4609	2967.4609	2360.0583	2360.0583
**Calculated monoisotopic mass [Da]**	2967.4324	2967.4324	2360.0419	2360.0532

Note: Bold text is used only to differentiate row and column headers for improved readability.

**Table 2 ijms-26-09962-t002:** The mass differences in b-type ions of sequences (3) and (4) of denosumab.

Dipeptide	Chemical Formula	Mass [Da]	Mass Difference [ppm]
**GT**	C_6_H_12_N_2_O_4_	176.0797	0.00
**SA**	C_6_H_12_N_2_O_4_	176.0797	
**b_2_(LE)**	C_11_H_18_N_2_O_4_H^+^	243.1340	−0.55
**b_2_(NK)**	C_10_H_18_N_4_O_3_H^+^	243.1452	−46.75
**b_2_ observed**	NA *	243.1338	- **
**b_3_(LEP)**	C_16_H_25_N_3_O_5_H^+^	340.1867	−3.52
**b_3_(NKP)**	C_15_H_25_N_5_O_4_H^+^	340.1979	−36.54
**b_3_ observed**	NA	340.1855	-
**b_4_(LEPE)**	C_21_H_32_N_4_O_8_H^+^	469.2293	−4.07
**b_4_(NKPE)**	C_20_H_32_N_6_O_7_H^+^	469.2405	19.87
**b_4_ observed**	NA	469.2312	-
**b_5_(LEPED)**	C_25_H_37_N_5_O_11_H^+^	584.2562	−12.44
**b_5_(NKPED)**	C_24_H_37_N_7_O_10_H^+^	584.2675	−6.78
**b_5_ observed**	NA	584.2602	-

* The chemical formula was not determined based on the observed m/z value of the ion. ** Denotes the reference mass used for comparison; therefore, the mass difference is not calculated for this entry.

**Table 3 ijms-26-09962-t003:** The comparison of MS parameters of the correctly identified peptide (sequence (6), variant B) and false-positive result (sequence (5), variant A).

Variant	A	B
**Sequence no.**	(5)	(6)
**Peptide sequence**	VCEFQFCNDPFLDVYYHK	VCEFQFCNDPFLGVYYHK
**Modification(s)**	D_succ, D_succ, (NEM, NEM)	Na+, (NEM, NEM)
**Mass accuracy [ppm]**	−1.56	−0.62
**Best ASR**	1.4	1.1
**Confidence score**	100	100
**Retention time [min]**	27.23	27.23
** *m* ** **/*z***	1241.0356	1241.0356
**Experimental monoisotopic mass [Da]**	2480.0566	2480.0566
**Calculated monoisotopic mass [Da]**	2480.0605	2480.0582

Note: Bold text is used only to differentiate row and column headers for improved readability.

**Table 4 ijms-26-09962-t004:** The comparison of MS parameters of the correctly identified peptide (sequence (8), variant D) and false-positive result (sequence (7), variant C).

Variant	C	D
**Sequence no.**	(7)	(8)
**Peptide sequence**	SYLTPVDSSSGWTAGAAAYYVGYLQPR	SYLTPGDSSSGWTAGAAAYYVGYLQPR
**Modification(s)**	Deamidation (Q25)	Carbamylation (S1)
**Mass accuracy [ppm]**	−7.98	0.75
**Best ASR**	1.6	1.1
**Confidence score**	100	100
**Retention time [min]**	27.62	27.62
** *m* ** **/*z***	1441.1763	1441.1763
**Experimental monoisotopic mass [Da]**	2880.3376	2880.3376
**Calculated monoisotopic mass [Da]**	2880.3606	2880.3355

Note: Bold text is used only to differentiate row and column headers for improved readability.

## Data Availability

The original contributions presented in this study are included in the article. Further inquiries can be directed to the corresponding author.

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
