# Peer review of "Peptide Mapping for Sequence Confirmation of Therapeutic Proteins and Recombinant Vaccine Antigens by High-Resolution Mass Spectrometry: Software Limitations, Pitfalls, and Lessons Learned"

_ijms, 2025, doi:10.3390/ijms26209962_

Round 1
Reviewer 1 Report
Comments and Suggestions for Authors
The manuscript by Dobrowolski et al. presents four examples where determining the respective masses of the investigated peptides was not enough to confirm the underlying amino acid sequences which were suggested from guiding sequence information. The guiding sequence information came from recombinant proteins whose amino acid sequences are known. The to be analyzed peptides were generated by enzymatic proteolysis. All four examples are taken from a larger selection which was investigated by mass spectrometric peptide mapping using up-to-date software tools; accumulated in the authors’ laboratory over several years. The manuscript is well written and should be of interest for mass spectrometry specialists working in the same research area as well as for beginners in the peptide mapping field.
Yet, there are a few issues which should be taken care of prior to publication:
1) The authors’ claim that the entireties of issues which come along with mass spectrometric peptide mapping are presented cannot be agreed to. The manuscript’s title: “The pitfalls of peptide mapping” is misleading in that sense and, hence, ought to be adapted to signal to the readers that the given limitation of the chosen examples to the topic is recognized by the authors. As there are four examples given, the correct title rather was: “Four pitfalls of peptide mapping”.
2) In section 3.2 the assumed differences seem of hypothetical nature. It remains unclear to the reader why the “false” sequences were selected to “almost” match with the experimentally determined peptide masses and/or fragment ion masses. The authors should provide more statistical information, e.g. FDRs, and more instrument precision-related numbers, such as rationalized cut-off values, to better explain the chosen setting.
3) The grammar needs to be checked throughout the text, best by a native speaker.
4) Each amino acid sequence which is shown in the manuscript (there are eight) should be labeled with unique numbers and the respective numbers should be used in the text, the tables, and the figures to clearly indicate which amino acid sequence is meant by the respective statements / entries.
Specific points (this list does not claim completeness):
Introduction
1) page 1, 2nd paragraph: the beginning sentence should be re-phrased to state that the correct amino acid sequence plays a crucial role for guaranteeing comparable biological activity as is seen with the original protein product.
2) page 1, second line from bottom: instead of “change” use “exchange”.
3) page 2, second paragraph: first sentence lacks a literature reference; please add.
4) page 2, third paragraph: “amino acid sequences of proteins”.
5) page 2, third paragraph: write “sequence analysis” not “sequence determination”.
Methods
1) page 2, last paragraph: “rRGB from SARS-CoV-2” ? please correct term and add to the list of abbreviations.
2) page 3, section 2.3: “The analyses were performed…”.
3) page 3, section 2.3: “ACN” not “Acn”.
Results
1) page 4, section 3.1, first paragraph: should start like: “With the 20 natural amino acids there are 190 possible …”.
2) Table 2 ff and Figures: Label the amino acid sequences with unique numbers.
3) page 6, section 3.2; title: The section’s sub-title needs to be changed to more precisely describe what the subject of the following section is. This reviewer understands that the authors give examples from the “selection” of potential PTMs which needs to be carefully chosen so that the software takes these options into consideration for calculating the theoretical masses of the (modified) peptides. As stated above, the selection criteria which chosen to start the investigations as thorough as done ought to be added.
4) page 6, section 3.2; first paragraph: the sentence: “Naturally, there are also high-abundance PTMs (e.g. pyroglutamine formation at the N-terminal of protein), that can occur in close to 100% content, and in that case the sequence can be confirmed with use of modified peptides.” needs to be re-phrased. It could read: “Naturally, there is also high-abundance PTMs (e.g. pyroglutamine formation at the N-termini of proteins). In those cases the sequences can be confirmed applying chemically synthesized modified peptides.”
5) page 6, section 3.2; second paragraph: the statement: “… the justification of found false positive was sought,…” ought to be re-phrased as it seems not logical to provide justification to a false positive finding.
6) page 6, section 3.2; last paragraph: the sentence: “… a sodium adduct, was also identified, and…” needs to be re-phrased since in the text the authors mention only the suggestions which were made by the software.
7) Figure 4: The figure caption should be corrected to remove lab jargon. The statement: “MS2 spectra of ys ion.” is very unconventional. The authors show MS2 spectra of defined precursor ions. Also, the color code needs to be explained.
Discussion
1) The text of the current discussion section rather resembles statements which form a conclusion. Thus, the headline of this section should be changed accordingly.
2) It is strongly recommended to add in a revised manuscript a real discussion section which contains a comparison of the points made in the results section with statements / findings of other publications which deal with sequence confirmation issues. Additionally, distinguishing remarks should be added to clearly separate this work’s statements from comparable and associated research fields, such as mass spectrometric de-novo sequencing (e.g. Yefremova et al., 2015; JASMS) to shed more light on scope and limitation of the presented work.
Comments on the Quality of English Language
see above
Author Response
Dear Reviewer,
Thank you very much for your thorough analysis of our manuscript and for all your valuable comments. We truly appreciate the time and effort you dedicated to providing such detailed feedback, which will help us improve the quality and clarity of the paper. Please find the detailed responses below and the corresponding revisions/corrections in track changes in the re-submitted file. In addition, following the guidance from the Editorial Office, the order of sections in the revised manuscript has been adjusted to: 1. Introduction; 2. Results; 3. Discussion; 4. Materials and Methods; 5. Conclusions.
Point-by-point response to Comments and Suggestions for Authors
Comments 1: The authors’ claim that the entireties of issues which come along with mass spectrometric peptide mapping are presented cannot be agreed to. The manuscript’s title: “The pitfalls of peptide mapping” is misleading in that sense and, hence, ought to be adapted to signal to the readers that the given limitation of the chosen examples to the topic is recognized by the authors. As there are four examples given, the correct title rather was: “Four pitfalls of peptide mapping”.
Response 1: We thank the reviewer for this insightful comment and fully agree that the original title might have implied a broader coverage than intended. We appreciate the suggestion and have revised the title to better reflect the scope of the manuscript. The new proposed title is: “Peptide Mapping for Protein Sequence Confirmation: Software Limitations, Pitfalls and Lessons Learned”
Comments 2: In section 3.2 the assumed differences seem of hypothetical nature. It remains unclear to the reader why the “false” sequences were selected to “almost” match with the experimentally determined peptide masses and/or fragment ion masses. The authors should provide more statistical information, e.g. FDRs, and more instrument precision-related numbers, such as rationalized cut-off values, to better explain the chosen setting.
Response 2: We appreciate the reviewer’s comment and recognize the concern regarding the potentially hypothetical nature of the assumed differences. In the revised version, we have expanded the Discussion section to address this uncertainty in greater detail. As stated in the Materials and Methods section (Table 5), our work relied solely on parameters directly available in the BioPharma Finder 5.1 software, without introducing external statistical algorithms. Importantly, in each example the “false” and “true” identifications were based on exactly the same raw data and processed with the same method, and the only difference was the reference sequence used for data interpretation. Therefore, the observed discrepancies cannot be attributed to a threshold setting but rather to the way the software handled fragment assignment. We believe that the extended discussion, together with the detailed description of parameters in Table 5, clarifies the rationale behind the presented examples.
Comments 3: The grammar needs to be checked throughout the text, best by a native speaker.
Response 3: We thank the reviewer for the helpful comments regarding language and grammar. We have carefully revised the manuscript to improve clarity and style, including the updated Discussion section and the newly added Conclusions. We kindly ask the reviewer to verify whether the revised version now meets the expected standard.
Comments 4: Each amino acid sequence which is shown in the manuscript (there are eight) should be labeled with unique numbers and the respective numbers should be used in the text, the tables, and the figures to clearly indicate which amino acid sequence is meant by the respective statements / entries.
Response 4: Unique numbers have now been assigned to each of the eight amino acid sequences, and these identifiers are consistently used throughout the text, tables, and figures to ensure clarity.
Comments 5: Introduction 1) page 1, 2nd paragraph: the beginning sentence should be re-phrased to state that the correct amino acid sequence plays a crucial role for guaranteeing comparable biological activity as is seen with the original protein product.
Response 5: We thank the reviewer for this valuable comment. In the first part of the paragraph, our intention was to emphasize from a biological perspective the crucial role of the amino acid sequence, highlighting its impact on biological activity, protein stability, and immunogenicity, supported by literature examples. In the following paragraph, we then pointed out that these correlations between sequence, function, stability, and immunogenicity form the basis for the requirements set by regulatory authorities.
Comments 6: 2) page 1, second line from bottom: instead of “change” use “exchange”.
Response 6: Following the suggestion, we revised the wording and used “substitution”, which we believe is most appropriate in the context of amino acid sequence alterations.
Comments 7: 3) page 2, second paragraph: first sentence lacks a literature reference; please add.
Response 7: Two relevant literature references have been added to the manuscript, namely:
Foltmann, B. Protein Sequencing: Past and Present. Biochem Educ 1981, 9, doi:10.1016/0307-4412(81)90049-2.
Gomes, A. V. On “A Method for the Determination of Amino Acid Sequence in Peptides” by P. Edman. Arch Biochem Biophys 2022, 726.
Comments 8: 4) page 2, third paragraph: “amino acid sequences of proteins”.
Response 8: The suggested change has been implemented.
Comments 9: 5) page 2, third paragraph: write “sequence analysis” not “sequence determination”.
Response 9: The suggested change has been implemented.
Comments 10: Methods 1) page 2, last paragraph: “rRGB from SARS-CoV-2” ? please correct term and add to the list of abbreviations.
Response 10: Our intention was to refer to the recombinant SARS-CoV-2 spike protein. The abbreviation “rS SARS-CoV-2” is used in our internal documentation; however, after checking the literature, we agree that the abbreviation “rS SARS-CoV-2” is not commonly used. To improve clarity, we now consistently use either “recombinant SARS-CoV-2 spike protein” or simply “SARS-CoV-2 spike protein” throughout the manuscript.
Comments 11: 2) page 3, section 2.3: “The analyses were performed…”.
Response 11: The suggested change has been implemented.
Comments 12: 3) page 3, section 2.3: “ACN” not “Acn”.
Response 12: The abbreviation has been corrected to “ACN” in the manuscript. The list of abbreviations has been updated accordingly.
Comments 13: Results 1) page 4, section 3.1, first paragraph: should start like: “With the 20 natural amino acids there are 190 possible …”.
Response 13: The suggested change has been implemented.
Comments 14: 2) Table 2 ff and Figures: Label the amino acid sequences with unique numbers.
Response 14: Unique numbers have now been assigned to each of the eight amino acid sequences, and these identifiers are consistently used throughout the text, tables, and figures to ensure clarity.
Comments 15: 3) page 6, section 3.2; title: The section’s sub-title needs to be changed to more precisely describe what the subject of the following section is. This reviewer understands that the authors give examples from the “selection” of potential PTMs which needs to be carefully chosen so that the software takes these options into consideration for calculating the theoretical masses of the (modified) peptides. As stated above, the selection criteria which chosen to start the investigations as thorough as done ought to be added.
Response 15: The intention of the sub-title was to emphasize that the software should not be allowed to introduce too many modifications within a single peptide, as this may lead to an artificial expansion of the search space and increase the risk of false positives. This point has now been further highlighted in the revised Discussion section. We kindly ask for the reviewer’s feedback on whether the current sub-title is now sufficiently clear, or if it still requires modification.
Comments 16: 4) page 6, section 3.2; first paragraph: the sentence: “Naturally, there are also high-abundance PTMs (e.g. pyroglutamine formation at the N-terminal of protein), that can occur in close to 100% content, and in that case the sequence can be confirmed with use of modified peptides.” needs to be re-phrased. It could read: “Naturally, there is also high-abundance PTMs (e.g. pyroglutamine formation at the N-termini of proteins). In those cases the sequences can be confirmed applying chemically synthesized modified peptides.”
Response 16: We have revised the sentence to improve clarity and scientific accuracy. Since PTMs (post-translational modifications) is a plural noun, the grammatically correct form is “there are” rather than “there is.”
Regarding the wording “modified peptides”, our intention was to indicate peptides that carry a specific post-translational modification (PTM). To avoid ambiguity, we have rephrased this part of the sentence and now use the term “peptides bearing the respective PTM” (or alternatively, “PTM-containing peptides”), which more precisely conveys the intended meaning. After the corrections, the text now reads: “Naturally, there are also high-abundance PTMs (e.g., pyroglutamine formation at the N-termini of proteins), which may reach nearly 100% occupancy. In those cases, the sequence can be confirmed by using PTM-containing peptides.”
Comments 17: 5) page 6, section 3.2; second paragraph: the statement: “… the justification of found false positive was sought,…” ought to be re-phrased as it seems not logical to provide justification to a false positive finding.
Response 17: The sentence has been revised for clarity and now reads: “If any false positives were detected, their underlying causes were investigated, and appropriate modifications to the evaluation method were introduced.”
Comments 18: 6) page 6, section 3.2; last paragraph: the sentence: “… a sodium adduct, was also identified, and…” needs to be re-phrased since in the text the authors mention only the suggestions which were made by the software.
Response 18: Throughout the manuscript we use the term identified consistently in the same sense as it is reported by the BioPharma Finder software, i.e., to denote assignments proposed by the algorithm, irrespective of whether they are correct or false. To maintain consistency, we did not replace this term only in the mentioned sentence. However, we have clarified the wording in this paragraph to explicitly state that the sodium adduct was identified by the software, which should avoid any potential misunderstanding. The sentence now reads “The same ion, while scanning against the correct variant (variant B), was identified by the software as a peptide with an aspartic acid to glycine substitution (Sequence (6)) and a sodium adduct, again with acceptable mass error due to the very similar masses of the full peptides, and a good ASR value.”
Comments 19: 7) Figure 4: The figure caption should be corrected to remove lab jargon. The statement: “MS2 spectra of ys ion.” is very unconventional. The authors show MS2 spectra of defined precursor ions. Also, the color code needs to be explained.
Response 19: The caption of Figure 4 has been revised to remove informal wording and to improve clarity. It now reads: “MS2 fragmentation spectra of the precursor ion at m/z 1441.68. Top: software-predicted spectrum; Bottom: experimental spectrum. Left: sequence (7), variant C; Right: sequence (8), variant D. The y8 fragment ion, highlighted in red, is shown in the zoomed region.”
Comments 20: Discussion 1) The text of the current discussion section rather resembles statements which form a conclusion. Thus, the headline of this section should be changed accordingly.
Response 20: In the revised version, the Discussion section has been substantially expanded, and a separate Conclusions section has been added. We kindly ask the reviewer to evaluate whether the revised structure now adequately addresses this concern.
Comments 21: 2) It is strongly recommended to add in a revised manuscript a real discussion section which contains a comparison of the points made in the results section with statements / findings of other publications which deal with sequence confirmation issues. Additionally, distinguishing remarks should be added to clearly separate this work’s statements from comparable and associated research fields, such as mass spectrometric de-novo sequencing (e.g. Yefremova et al., 2015; JASMS) to shed more light on scope and limitation of the presented work.
Response 21: In the revised version, the Discussion section has been substantially expanded and now includes comparisons between our observations and statements from other publications addressing sequence confirmation. Furthermore, we have added distinguishing remarks to clearly separate our work from related research areas, such as mass spectrometric de novo sequencing (e.g., Yefremova et al., 2015; JASMS), in order to better highlight the scope and limitations of the presented study. We kindly ask the reviewer to verify whether the revised Discussion section now meets these expectations.
Reviewer 2 Report
Comments and Suggestions for Authors
In the current manuscript, the authors discuss the challenges associated with peptide mapping mass spectrometry. However, these challenges are well-documented in existing literature, and a range of peptide mapping algorithms have already been developed or are under active development. The isobaric dipeptide (SA and GT) examples referenced in the manuscript can also be distinguished using alternative modern fragmentation methods, including but not limited to photodissociation, electron transfer dissociation (ETD), or electron activated dissociation (EAD). Overall, the manuscript does not present a systematic investigation of the topic and does not present novel findings or contribute additional insights to the existing body of knowledge.
I have listed a few major comments that may help to improve the manuscript.
- The title needs to be specific.
- The abstract should be updated to incorporate the experimental findings.
- The material and methods section lacks specific information. Sample preparation is too vague, and there are no specific details mentioned. The authors should provide specific details or cite a reference. Sample analysis and data analysis should also have more details.
- Table 1 does not provide any new information.
- The results section needs a thorough revision as it is not systematically outlined and does not provide specific evidence. The Dipeptides with similar, or identical mass, the 100ppm statements are not relevant.
- The subsection The posttranslational modification – you can(‘t) have too many, is hard to follow, and the authors are not clear what they want to demonstrate.
Author Response
Dear Reviewer,
We sincerely thank the reviewer for the feedback and for highlighting important aspects of the topic. In the revised version of the manuscript, the Introduction has been updated to more clearly define the aim of the work, which is to highlight practical challenges in residue-by-residue sequence confirmation using commercial software, where the analysis relies on built-in evaluation parameters (e.g., confidence score). The scope of the manuscript is focused on a regulated environment, where academic algorithms, although highly valuable in research, are not directly applicable due to the requirement for software validation in GxP settings.
Regarding the isobaric dipeptide examples, we agree with the reviewer that these can indeed be distinguished using advanced fragmentation approaches such as ETD, EAD, or photodissociation, and even by simpler strategies, such as analyzing synthetic peptides and comparing their retention times on RP columns. In our manuscript, we illustrated that even with HCD fragmentation it was possible to differentiate the sequences: in one case no fragmentation was observed between serine and alanine, whereas clear fragmentation was detected between glycine and threonine, which allowed us to confirm that sequence (1) was correct. In this example, the point was not to identify SA or GT as isolated dipeptides, since they constitute an integral part of a longer peptide. Rather, it was to demonstrate that relying solely on software-provided parameters such as ASR and confidence score would not have been sufficient to unequivocally exclude sequence (2) if it had been used as the reference sequence.
We kindly ask the reviewer to re-evaluate the revised manuscript, particularly considering the clarified Introduction, the substantially expanded Discussion, and the newly added Conclusions. Please find the detailed responses below and the corresponding revisions/corrections in track changes in the re-submitted file. In addition, following the guidance from the Editorial Office, the order of sections in the revised manuscript has been adjusted to: 1. Introduction; 2. Results; 3. Discussion; 4. Materials and Methods; 5. Conclusions.
Point-by-point response to Comments and Suggestions for Authors
Comments 1: The title needs to be specific.
Response 1: We thank the reviewer for this comment and fully agree that the original title might have implied a broader coverage than intended. We appreciate the suggestion and have revised the title to better reflect the scope of the manuscript. The new proposed title is: “Peptide Mapping for Protein Sequence Confirmation: Software Limitations, Pitfalls and Lessons Learned”
Comments 2: The abstract should be updated to incorporate the experimental findings.
Response 2: The abstract has been revised to more explicitly highlight the observations and conclusions derived from the presented examples, in order to better reflect the experimental findings of the study.
Comments 3: The material and methods section lacks specific information. Sample preparation is too vague, and there are no specific details mentioned. The authors should provide specific details or cite a reference. Sample analysis and data analysis should also have more details.
Response 3: The focus of our manuscript is not on sample preparation itself but rather on the challenges associated with data interpretation during residue-by-residue sequence confirmation. For this reason, the Materials and Methods section was intentionally kept at a general level. To guide interested readers, in the Introduction we have included references that describe sample preparation and peptide mapping protocols in detail (e.g., Lam et al., 2022, doi:10.1016/j.omtm.2022.09.008; Formolo et al., 2015; doi: 10.1021/bk-2015-1201.ch001, Mouchahoir & Schiel, 2018, doi:10.1007/s00216-018-0848-6).
In each example presented in our manuscript, the observed differences arose from the reference sequence used for data analysis, while the raw data correspond to the same LC-MS/MS run of a single sample. This means that sample preparation could not have influenced the misidentifications, as in each case the same precursor ion was present but assigned differently by the software. We would kindly appreciate clarification from the reviewer regarding which specific experimental details they consider important, so that we can further improve the transparency of the Materials and Methods section.
Comments 4: Table 1 does not provide any new information.
Response 4: After the re-organization of sections following the Editorial Office guidance, the numbering has changed and the table in question now appears as Table 5. This table summarizes the parameters provided by the BioPharma Finder software. As some of these parameters are specific to this platform (e.g., confidence score, ASR), we believe that presenting them in tabular form is helpful for the clarity and completeness of the manuscript. We also understand that the reviewer has pointed out a need for more detail in the Materials and Methods section, and we consider Table 5 an important element in addressing that concern.
Comments 5: The results section needs a thorough revision as it is not systematically outlined and does not provide specific evidence. The Dipeptides with similar, or identical mass, the 100ppm statements are not relevant.
Response 5: We thank the reviewer for this comment. Due to its general nature, it is somewhat difficult for us to provide a specific response, and we would kindly appreciate further clarification on which aspects of the Results section the reviewer considers most problematic. We would like to note that the revised Introduction now explicitly states that the presented examples are based on the use of commercial software and its built-in evaluation parameters, which should make the Results section easier to follow. These aspects are further elaborated in the expanded Discussion to reinforce the overall aim of the manuscript.
As an example, the issue of the 100 ppm error for dipeptides illustrates that when a dipeptide is a part of a larger peptide, the relative mass error rapidly decreases to an acceptable level (refer to table 2). In our case, if only the incorrect sequence (4) had been used as the reference, the ASR value (1.2) and the confidence score (100) alone could have led to acceptance of this peptide. Only by carefully examining which b and y ions were present could one recognize that a specific fragment of the sequence was not actually covered. In the revised manuscript, each sequence has now been assigned an individual number, which we hope will make this part easier to follow.
We hope that the changes introduced have improved the clarity of this section, and if the reviewer considers that further modifications are needed, we would greatly appreciate detailed feedback where possible so that we can address them accordingly.
Comments 6: The subsection The posttranslational modification – you can(‘t) have too many, is hard to follow, and the authors are not clear what they want to demonstrate.
Response 6: We thank the reviewer for this remark. As addressed in the revised manuscript, the Introduction has been clarified, the Discussion expanded, and individual numbering introduced for each sequence. We hope these changes have made this subsection easier to follow and its purpose clearer. If there are still specific points that remain unclear, we would greatly appreciate more detailed feedback so that we can address them accordingly.
Round 2
Reviewer 2 Report
Comments and Suggestions for Authors
The revised version shows limited improvement. The authors addressed only minor corrections, did not specifically revise the subsections, and did not provide sufficient supporting evidence.
- The abstract resembles an introduction and does not clearly present any specific scientific findings or conclusions. The authors mention, “Drawing from extensive practical experience, this work provides practical observations and best practices aimed at residue-by-residue sequence confirmation.” However, readers generally expect to see concrete results highlighted in the abstract.
- The authors did not revise the sample preparation and not cited any reference. It is OK to keep the sample preparation at general level but cite the reference. The author stated “ The standard bottom-up approach with several fit-for-purpose modifications was used. In developed standard operating procedure the studied protein (100-125 µg) was first denatured using high concentration of chaotropic salt (6 M, guanidine hydrochloride), followed by reduction of disulfide bridges with TCEP (Tris(2-carboxyethyl)phosphine hydrochloride). Reassembly of disulfide bridges was prevented by alkylation of free cysteines with NEM (N-ethylmaleimide). Then, the guanidine hydrochloride and remaining reagents were removed from the solution via ultrafiltration. The digestion was performed under mild denaturing conditions (1 M urea) using trypsin or chymotrypsin.” Which are the several fit-for purpose modifications? What is developed standard operating procedure? Cite the reference of this SOP. What is specific concentration of TCEP, NEM, trypsin or chymotrypsin? How long the proteins were digested and at what temp/pressure?
- Table 1 and currently table 5 still not provide any new information.
- The results section still needs thorough revision. The first two paragraphs of the subsection- 2.1. Dipeptides with similar, or identical mass - is introduction and 100 ppm mass difference is irrelevant in results section. Also, the remaining paragraph is observation without actual spectra. The authors should provide the actual spectra and indicate the similar ions with their mass errors compared with the figure 1/table 2 fragmentation pattern to pinpoint the discrepancy in peak assignment and fragmentation pattern.
- The subsection- “ 2.2. The post-translational modification – you can(‘t) have too many” still difficult to understand. First paragraph is history. Suddenly second paragraph is SARS-CoV-2 spike protein. Subsection 2.1 is about the mAb and 2.2 is about SARS-CoV-2 spike protein and in 2.2 third paragraph is again stated “In the first example,…” which one? What is the source of sequence 5 and 6? The authors should state it in the methods section.
- The authors stated “Good coverage with b and/or y ions of the shared regions of the peptides was observed,” provide the spectra with the sequence coverage and fragmentation pattern. Similar comments for sequence 7 and 8.
Author Response
Dear Reviewer,
We sincerely thank the reviewer for all the constructive comments. Following the suggestions, the manuscript has been supplemented with mass spectra to better illustrate the sequence coverage and fragmentation patterns. For each of the four examples, we have included two sets of spectra (one for each compared sequence). Each set consists of a software-predicted spectrum and the corresponding experimental spectrum, allowing the reader to directly compare the expected m/z values of the b and y ions with those observed experimentally. In total, eight figures have been added. We are, however, slightly concerned that including this number of figures in the main text may make the manuscript more difficult to follow. We would therefore kindly ask whether the reviewer considers it essential to retain all of them in the manuscript, or whether it might be more appropriate to present only selected representative examples in the main text and move the remaining spectra to the Supplementary Materials.
Comments 1: The abstract resembles an introduction and does not clearly present any specific scientific findings or conclusions. The authors mention, “Drawing from extensive practical experience, this work provides practical observations and best practices aimed at residue-by-residue sequence confirmation.” However, readers generally expect to see concrete results highlighted in the abstract.
Response 1: We appreciate the Reviewer’s comment regarding the lack of specific scientific findings in the original abstract. In response, we have revised the abstract to clearly reflect the key examples discussed in the Results section. These include misassignments due to isobaric dipeptides in antibody analysis and software-induced artifacts in SARS-CoV-2 spike protein mapping. The revised version now emphasizes concrete observations and better communicates the practical implications of our study.
Comments 2: The authors did not revise the sample preparation and not cited any reference. It is OK to keep the sample preparation at general level but cite the reference. The author stated “ The standard bottom-up approach with several fit-for-purpose modifications was used. In developed standard operating procedure the studied protein (100-125 µg) was first denatured using high concentration of chaotropic salt (6 M, guanidine hydrochloride), followed by reduction of disulfide bridges with TCEP (Tris(2-carboxyethyl)phosphine hydrochloride). Reassembly of disulfide bridges was prevented by alkylation of free cysteines with NEM (N-ethylmaleimide). Then, the guanidine hydrochloride and remaining reagents were removed from the solution via ultrafiltration. The digestion was performed under mild denaturing conditions (1 M urea) using trypsin or chymotrypsin.” Which are the several fit-for purpose modifications? What is developed standard operating procedure? Cite the reference of this SOP. What is specific concentration of TCEP, NEM, trypsin or chymotrypsin? How long the proteins were digested and at what temp/pressure?
Response 2: In the revised version, a literature reference has been added to support the general description of the standard operating procedure. Furthermore, we have supplemented the section with the specific concentrations of the reagents used (TCEP, NEM, trypsin, and chymotrypsin) as well as the detailed conditions for enzymatic digestion with trypsin and chymotrypsin (time and temperature). We hope that these additions clarify the methodology and address the reviewer’s concerns.
Comments 3: Table 1 and currently table 5 still not provide any new information.
Response 3: In the revised version, Table 5 has been removed, and the description of the data analysis workflow has been expanded directly in the text. In particular, we have added the definitions of two software-specific parameters of BioPharma Finder that were used for peptide identification in this work, namely the confidence score and the Average Structural Resolution (ASR), to clarify their meaning and role in the analysis, especially for readers who may not be familiar with this specific software.
Comments 4: The results section still needs thorough revision. The first two paragraphs of the subsection- 2.1. Dipeptides with similar, or identical mass - is introduction and 100 ppm mass difference is irrelevant in results section. Also, the remaining paragraph is observation without actual spectra. The authors should provide the actual spectra and indicate the similar ions with their mass errors compared with the figure 1/table 2 fragmentation pattern to pinpoint the discrepancy in peak assignment and fragmentation pattern.
Response 4: The descriptive paragraphs in subsection 2.1 were intended as a necessary introduction to demonstrate that the presented examples originate from real situations encountered by the authors. We understand the reviewer’s concern that such content may not fully fit within the Results section, and therefore the Introduction has been expanded to provide the reader with the appropriate background. In addition, mass spectra have now been included for all examples, showing the m/z range that illustrates the key differences in the fragmentation pattern, thereby pinpointing the discrepancies in peak assignment.
Comments 5: The subsection- “ 2.2. The post-translational modification – you can(‘t) have too many” still difficult to understand. First paragraph is history. Suddenly second paragraph is SARS-CoV-2 spike protein. Subsection 2.1 is about the mAb and 2.2 is about SARS-CoV-2 spike protein and in 2.2 third paragraph is again stated “In the first example,…” which one? What is the source of sequence 5 and 6? The authors should state it in the methods section.
Response 5: The first paragraph of subsection 2.2 has now been moved to the Introduction, where we additionally emphasized that the results presented in subsection 2.1 are derived from our work with monoclonal antibodies (denosumab), while subsection 2.2 provides examples from our work with the SARS-CoV-2 spike protein. Our laboratory routinely works with different classes of therapeutic proteins, including vaccine antigens, and the examples were selected to reflect this broader experience. In addition, we revised the ambiguous wording “in the first example” so that it now reads “In the first example of false positives results search (…)” to avoid confusion. We hope that these changes will make it easier for the reader to understand that the subsections present different types of case studies.
Comments 6: The authors stated “Good coverage with b and/or y ions of the shared regions of the peptides was observed,” provide the spectra with the sequence coverage and fragmentation pattern. Similar comments for sequence 7 and 8.
Response 6: In line with the earlier points, mass spectra have now been added to the manuscript for the relevant examples. These spectra illustrate the sequence coverage and fragmentation patterns, and in particular highlight the differences in the presence of specific fragment ions for the regions where the peptide sequences differ.